# Measuring the influence of depressive symptoms on engagement, adherence, and weight loss in an eHealth intervention

**Lex Hurley** [1]*, **Nisha G. O'Shea**[2], **Julianne Power**[1], **Christopher Sciamanna**[3], **Deborah F. Tate**[1,4,5]

**1** The University of North Carolina at Chapel Hill, Department of Health Behavior, Chapel Hill, North Carolina, United States of America, **2** Research Triangle Institute (RTI) International, Research Triangle Park, North Carolina, United States of America, **3** Penn State Cancer Institute, Department of Medicine, Hershey, Pennsylvania, United States of America, **4** The University of North Carolina at Chapel Hill, Department of Nutrition, Chapel Hill, North Carolina, United States of America, **5** Nutrition Research Institute, University of North Carolina at Chapel Hill, Kannapolis, North Carolina, United States of America

* lexh@unc.edu

## Abstract

### Background

Digital behavior change interventions (eHealth, mHealth) are known to be capable of promoting clinically significant weight loss among some participants. However, these programs can struggle with declining engagement and adherence over time, which can hamper their effectiveness. This analysis examines the extent that depression symptoms may negatively influence engagement, adherence, and 6 month weight change in an eHealth intervention.

### Methods

Structural equation modeling is applied to test the effects of baseline depression symptoms on weight change outcomes, mediated through latent constructs of engagement and adherence, respectively. These constructs were highly correlated within this dataset and necessitated two separate models to be tested. Engagement was indicated by 6 month sums of website logins, user-created goals, visiting various webpages, and posts on the online discussion boards. Adherence was indicated by 6 month sums of weeks exercise goals met, days weight logged, and days of complete dietary tracking.

### Results

Depression symptoms showed no direct association with weight change (p's ≥ 0.6), but were negatively associated with both constructs of engagement and adherence (p's < 0.001), which in turn were negatively associated with weight change in both models (p's < 0.001). It was determined depression symptoms had a positive indirect association with weight change fully mediated through these variables, meaning less weight loss or possible weight gain (p < 0.001).

**Data availability statement:** The data used in this analysis is hosted on the Carolina Digital Repository from UNC Chapel Hill Libraries, and may be found at the following link: https://cdr.lib.unc.edu/concern/data_sets/th83mc93g?locale=en

**Funding:** This research was supported by a Cancer Health Disparities training grant T32CA128582 (awarded to: LH) from the National Cancer Institute (www.cancer.gov). The parent study was funded by the National Institute of Diabetes and Digestive and Kidney Diseases (www.niddk.nih.gov) R01DK095078 (awarded to: DT, CS). The funders had no role in study design, data collection and analysis, decision to publish, or preparation of the manuscript.

**Competing interests:** The authors have declared that no competing interests exist.

## Discussion

This analysis shows that depression symptoms had a significant, undesirable effect on weight loss outcomes within this eHealth intervention, fully mediated through measured participant engagement and adherence. Further research is needed to test these constructs within a longitudinal model to better understand their dynamic interrelationships, and consider means to address depression in future digital interventions.

## Author summary

Digital behavior change interventions (DBCIs) have shown promise for developing scalable interventions to help address health outcomes such as overweight and obesity. However, it is known that engagement with DBCIs tends to drop off before an intervention is finished, thus reducing its potential for effectiveness. Depression is often mentioned as a likely contributor to reduced engagement and/or adherence in these programs, but is rarely studied in-depth outside of mental health-specific interventions. We applied structural equation modeling to test the extent that depression may influence weight change outcomes directly, or mediated through latent constructs of engagement and adherence to try and solidify the literature. We found that depression symptoms were fully mediated through these constructs; meaning that higher levels of depression had negative impacts on engagement and adherence, which in turn have negative impacts on weight (i.e., more weight loss). We concluded that depression symptoms have a significant indirect positive effect on weight, meaning less weight loss or possible weight gain. Our study provides a basis to consider baseline depression levels as an important variable to consider when designing future DBCIs for weight management, and possibly other outcomes.

## Introduction

Digital behavior change interventions (DBCIs) are able to benefit a variety of health outcomes, including weight management, physical activity promotion, as well as smoking and other substance use cessation; often eliciting equal or greater effect sizes than traditional in-person interventions [1–5]. These programs feature increased scalability to reach multitudes of participants for lower cost than in-person programs and are largely accessible at participants' convenience. As over 70% of U.S. adults are living with overweight or obesity, which increase risks of numerous chronic diseases, certain types of cancers, and other morbidities, this improved scalability is a key asset to support contemporary weight management interventions [6,7]. While the literature is robust with recommendations and resources to design and implement DBCIs for a variety of health outcomes, their effectiveness inevitably depends on how much participants use these programs. This analysis investigates one potential issue which may be contributing to reduced participant engagement, adherence, and success in DBCIs.

Participant engagement and adherence are both necessary precursors to DBCI effectiveness, which happen more or less in tandem during an intervention. Despite their importance to DBCI research, definitions of these constructs have tended to vary between publications [8]. It is clear that engagement and adherence are inextricably linked process; however, they are conceptually and empirically distinct. Definitions and operationalizations are provided here to promote clarity and comparability.

*Engagement* is defined here as the extent to which a user interacts with the digital interface of an intervention as intended, based on the systematic definition proposed by Perski *et al* [8]. Engagement is a latent construct that may be measured using objective usage indicators including logins, page views, proportion of full program components utilized, as well as messages, resources or lessons read [9,10]. Engagement with DBCIs is typically considered essential to promote the dose received of intervention content and is a common focus of DBCI research. There is an often-cited 'Law of Attrition' which broadly states since DBCIs operate at the discretion of participants who may choose to stop using them, they suffer from rapid declines in user engagement, increased risks of disengagement (i.e., all usage indicators dropping to zero, but not lost to follow-up; a.k.a. "non-usage attrition"), and possible attrition or loss to follow-up [11–15]. There are several proposed hypotheses explaining these observed high disengagement rates, including replacement discontinuance, where users may switch to some preferred alternative to the DBCI, and disenchantment discontinuance, where users may disengage from a program because they are dissatisfied with it in some way [12]. While much DBCI research focuses on increasing or sustaining program engagement among active participants, comparatively less research investigates factors which may negatively influence participant engagement and adherence, and possibly contribute to this hypothesized replacement or disenchantment discontinuance mechanisms.

*Adherence* is the next step along the pathway from engagement to successful intervention outcomes, and is defined here as the degree that a participant's actual (measured) behavior corresponds with requests or goals from an intervention. This definition of adherence is in line with the WHO, and is generally similar to definitions used in other contemporary publications. [16–20] In weight management programs, these recommendations would typically include adherence to dietary regimens [21], physical activity goals [22], daily weighing recommendations, attendance to counseling sessions, and regular self-monitoring of behaviors and outcomes [23,24]; to contribute to improved weight outcomes [25–27].

It is well known that depression and weight status are strongly associated such that individuals of higher bodyweight tend to also exhibit higher depressive symptoms, and that depressive symptoms can improve with successful weight loss in traditional in-person lifestyle modification interventions (not considering very low bodyweights) [28–31]. Based on research in traditional in-person interventions, baseline depressive symptoms have been associated with increased attrition and fewer intervention sessions attended [28,32]. There is some evidence of a mediating pathway, as Wing, Phelan, and Tate found that baseline depression symptoms negatively influenced mediating program adherence, and thus weight change within an in-person intervention [25]; however, baseline depressive symptoms do not often predict direct variance in weight change [29,33,34]. As weight-related stigma is a potential factor influencing adherence to these in-person programs [28], remote DBCIs may be a promising alternative due to a lower threshold of interaction to attend sessions and lower perceived risk of encountering stigma.

Unfortunately, depression and poor mental health are not often studied in DBCIs outside of those specifically focusing on mental health outcomes, and rarely quantified in depth. In their systematic review, Perski *et al* identify several studies which mention isolated associations between mental health and differential adherence or engagement in DBCIs, and possibly signaling increased risk of participant dropout [8,18,35–44]. Additionally, in their systematic review of factors influencing adherence in DBCIs for noncommunicable diseases, Jakob *et al* found that depression symptoms were reported to be negatively associated with adherence for several outcomes including physical activity and weight loss, and only positively associated with adherence in DBCIs specifically targeting depression [20]. As higher levels of depression are known to negatively influence motivation and increase feelings of fatigue and

selective focus on negative information [28], which could increase the risk of factors such as disenchantment discontinuance, it is valuable to measure how depression may influence the constructs of engagement and adherence in DBCIs. Therefore, this analysis seeks to test how depression symptoms may influence distal weight change outcomes when modeled through engagement and adherence as mediators.

The guiding hypotheses for this analysis are: 1) depression symptoms will have a negative influence on the latent constructs of engagement and adherence; 2) engagement and adherence will each be negatively associated with 6-month weight change (i.e., weight loss); and 3) the direct effect of depressive symptoms on weight change will be fully mediated by program engagement and adherence.

## Materials and methods

### Dataset

Data for this analysis come from LoseNow PA (LNPA), an NIH-funded 12-month cluster-randomized controlled eHealth DBCI for weight management among primary care patients living with overweight or obesity in the United States (clinicaltrials.gov identifier: NCT01606813) [45]. In this study, primary care providers (PCPs; k = 31) were randomly assigned to one of three intervention groups, and patients (N = 550) received the intervention assigned to their provider: 1) enhanced usual care (EUC; n = 187); 2) internet weight loss intervention (IWL; n = 181); or 3) internet weight loss intervention plus automated physician-tailored advice (IWL+PCP; n = 182) [45]. Summarizing primary outcomes of the trial, both the IWL and IWL+PCP arms exhibited significantly greater weight loss than EUC, and there were no significant differences between IWL and IWL+PCP on any outcomes. Full details of the trial can be found in Tate *et al.*, 2022 [45].

Both IWL intervention arms had access to a study website requiring a username and password to log in. The study website included instructional lessons and resources; self-monitoring diary pages for diet, physical activity, and weight; weekly computer-tailored feedback messages and graphs; personal goal setting and problem-solving tools; progress summary pages; a social forum to message with other participants; as well as opt-in reminder text messages containing updates, encouragement, and motivational content. Participants could choose from several Eating and Activity Plan monitoring options to customize how they monitored calories and how exercise goals would progress, respectively, and shift their monitoring plan(s) if desired during the intervention. The primary difference between the intervention arms was that participants in the IWL+PCP condition also received brief biweekly emails containing a computer-generated tailored message addressed from their PCP related to weight loss progress, frequency of website log-ins, time in the program, and other aspects such as patient-reported motivation. PCP's could view and edit these messages for each patient before sending, though results showed that this functionality was rarely used by PCPs (only 1.2% of all PCP messages sent were edited) [45].

Since the study website was functionally identical across both intervention arms and no significant influence of the automated physician advice messaging was found, participants from both arms were pooled to maximize the statistical power of this secondary analysis. Of the full N = 363 participants assigned to the IWL and IWL+PCP groups, 24 did not ever log in to the website and are excluded from this analysis, bringing the effective sample size to n = 339. Additionally, this analysis specifically focuses on the first 6 program months to best measure the effects of active program engagement and adherence with the following rationale: This timeframe is critical as most DBCI engagement occurs in the early months of an intervention, most weight loss occurs during months 1-6, and users are increasingly likely to disengage from DBCIs as duration increases in months 6-12 [12,46–48].

### Ethics statement

The parent LNPA study was approved by institutional review boards at the University of North Carolina at Chapel Hill (#12-1661) and Penn State College of Medicine (#39237). Both PCPs and patients completed written informed consent forms prior to data collection and randomization, as described in Tate *et al*, 2022 [45].

### Measures

LNPA participants completed study assessments at baseline, 3, 6, and 12 months. During baseline assessments, participants self-reported various sociodemographic characteristics including age, sex, and race/ethnicity, which are applied as exogenous covariates. Weight change is measured by a difference score between participant weights at baseline and 6 months, and is the primary dependent variable of analysis. Sensitivity analyses using 12-month weight change data, as well as percent weight change at 6 and 12 months, were applied for added confidence in estimates.

The Centers for Epidemiology – Depression (CES-D) scale was administered as part of baseline study assessments and is a validated measure of depression symptoms [49]. Briefly, the CES-D is a non-diagnostic psychometric scale with score values ranging from 0-60, with higher values indicating stronger depression symptoms, and scores ≥16 commonly referenced as indicating a risk of clinical depression [49]. Standardized CES-D scores were used as the primary exogenous predictor of interest in the following simultaneous equation models.

Indicators of engagement with the IWL study website include logins, page hits, personal goals created, and social forum posts – all of which have been previously validated as indicators for this construct [9]. Website page hits refer to the number of time-stamped website URLs visited by logged user IDs. To reduce collinearity with other indicators, the page hits variable only encompasses URLs that did not include the login page, home page, any of the behavioral self-monitoring pages, or the contact/help page. Therefore, increasing values of this variable indicate users accessing more lessons, resources, viewing feedback pages, etc. for a greater breadth of program usage. On the first website login each week, participants could create and track up to 3 personal goals, which were summed as an additional engagement indicator after being found to be associated with weight loss in a previous analysis [50]. The social forum posts variable is a sum of the number of posts users made on the IWL message boards where participants could communicate with other participants and LNPA staff if desired.

Indicators of adherence with LNPA program recommendations include the number of weeks the minimum physical activity goals were met, days of complete dietary logging, as well as days bodyweight was logged, and were operationalized as follows: The number of weeks that users logged a number of active minutes of exercise that met or exceeded the lowest intensity Activity Plan goals for that point in time were summed to indicate meeting minimum program goals for that week. The number of days participants either directly logged or adjusted dietary entries to include ≥ 800 calories were summed to indicate complete days of dietary monitoring, based on previous literature which indicated that this much logging is associated with improved weight management [51,52]. Lastly, days that users self-monitored their bodyweight were summed to demonstrate additional adherence to program recommendations, as the website had dedicated pages and feedback graphs for tracking weight change over time.

### Data analysis

**Missing data.** At 6 months, 26.7% of participants were missing weight measurements, and 17.6% were missing 12-month weight measurements. No clear patterns of missingness

were discernible. Assuming these observations are missing at random, we applied multiple imputation using chained equations to pool point and standard error estimates across m = 50 iterations, including the most recent participant weight logged on the IWL website before 6 months in the dataset to facilitate weight imputations [53–59].

**Structural equation modeling.** We used a structural equation modeling (SEM) approach to efficiently model mediation pathways between engagement and adherence as latent variables (LVs), and applied log-link transformations of indicator variables where appropriate. LVs of engagement and adherence were separately examined using confirmatory factor analysis to determine optimal loading variables: sum logins for engagement and sum weeks PA goals met for adherence (indicated by the "*" symbols in Fig 1). We first conducted measurement models of the adherence and engagement LVs, comparing the fit of a 1- and 2-factor model. During this phase, we found that the two LVs were highly correlated with one-another (>95%) when entered into the same model. Because of this, it was deemed necessary to estimate two separate mediation models to test study hypotheses. Path diagrams of these twin models are displayed in Fig 1.

Model fit was assessed using Chi-square ($\chi^2$) fit test, root mean square error of approximation (RMSEA), confirmatory factor index (CFI), and Tucker Lewis index (TLI) values, as well as standardized root mean square residuals (SRMR). All analyses were conducted using the *lavaan*, *semTools*, and *mice* packages for R statistical software [59,56].

## Results

### Descriptive statistics

Most LNPA participants randomized to intervention arms (N = 363) were white (82.3%) and female (70.3%) with a mean baseline weight of 97.99 kg (216.03 lbs) and average age 51.86

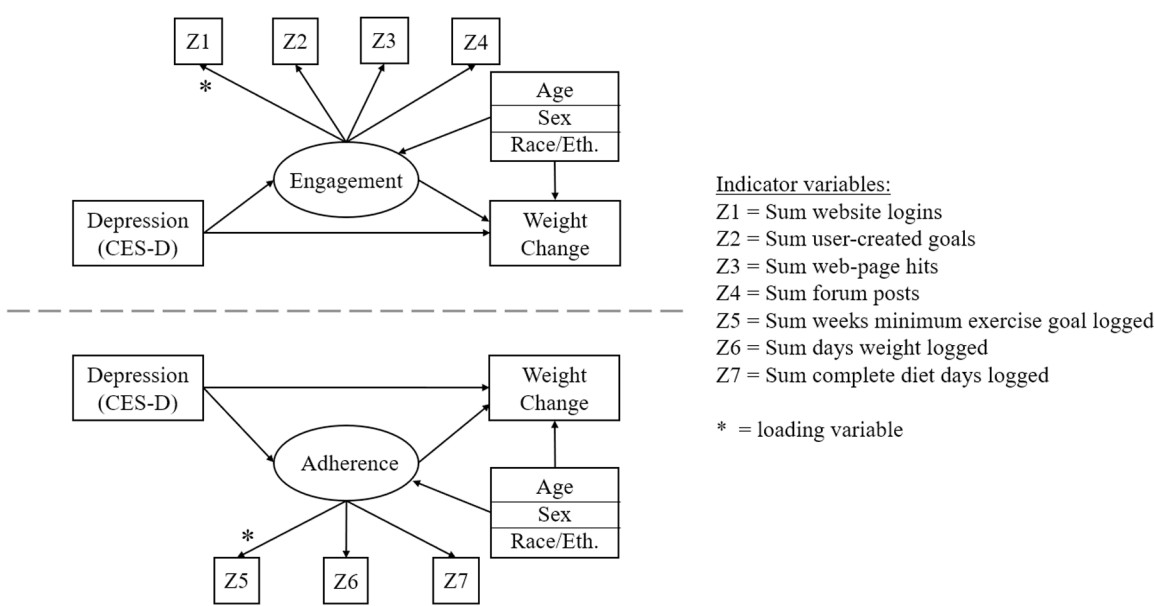

**Fig 1. Structural equation model path diagrams.** Describing the paths of these SEMs, each model estimates the direct effects of exogenous baseline depression symptoms measured by the CES-D on 6-month weight change, as well as on endogenous LVs of engagement and adherence, which in turn each have a direct effect mapped onto 6-month weight change. Three separate exogenous variables of participant age, sex, and race/ethnicity are mapped onto adherence, engagement, and weight change as demographic control covariates.

years. Over half of the sample (~56.7%) reported having less than a Bachelor's degree or equivalent education. Average CES-D scores at baseline were 13.13 and ranged between 0-41, with 112 (30.9%) participants reporting scores above the threshold of ≥16 indicating a risk of clinical depression. A full descriptive summary of baseline characteristics for those with data is displayed in Table 1. Two participants were identified as outliers with excessive weight losses in this sample; however, their inclusion did not meaningfully influence model results, so they remained in this analysis. There were no significant differences in any variables between the full and subset samples.

Website engagement indicators exhibited high variability. Approximately 53 participants (14.6%) logged in 5 times or fewer over 6 months, while there were several users with extremely high engagement indicators (e.g., one user had 977 website logins and 4,446 registered page views within 6 months, compared to the sample average of approximately 99 logins and 300 page views). This high variance in engagement indicators persisted even after outlier testing and removal. Adherence indicators exhibited much lower variability by comparison. A correlation matrix of all indicator variables used in this analysis are available in S1 File. Descriptive statistics for all indicator variables are summarized in Table 2.

## Simultaneous equation model fit and estimates

Both models converged across m = 50 imputed datasets with good fit statistics overall. The chi square ($\chi^2$) test statistics for engagement were both significant (p's ≤ 0.001) which is undesirable, yet not surprising as the $\chi^2$ tests the null hypothesis that a given model fits the data perfectly, and with a modest sample size of n=339, even small deviations could produce significant values [53]. Pooled RMSEAs for engagement = 0.063 (p = 0.19) and adherence = 0.089 (p = 0.02) indicate that both models fit the data fairly well, though the engagement model shows better fit. Pooled SRMRs for engagement = 0.033 and adherence = 0.023 which

**Table 1. Participant characteristics of pooled intervention groups (N = 363).**

| Variable | Frequency (%) | Mean (SD) | Min; Max |
|---|---|---|---|
| **Gender** | | | |
| Female | 255 (70.3%) | | |
| Male | 93 (25.6%) | | |
| **Race** | | | |
| White | 299 (82.3%) | | |
| Black | 45 (12.5%) | | |
| Other POC | 19 (5.2%) | | |
| **Highest Education Achieved** | | | |
| High School | 89 (24.5%) | | |
| 1-3 Years College | 117 (32.2%) | | |
| 4+ Years College | 139 (38.3%) | | |
| **Age** | | 51.86 (10.86) | 21; 70 |
| **Clinic Weight, Baseline** (kg) | | 97.99 (18.73) | 62.1; 148.6 |
| **Clinic Weight, 6 months** (kg) | | 92.63 (18.23) | 57.6; 157.75 |
| **Clinic Weight, 12 months** (kg) | | 93.61 (18.72) | 58.2; 167.60 |
| **6-month Weight Change** (kg) | | -4.72 (5.79) | -30.0; +6.9 |
| **CES-D Scores** (0-60) | | 13.13 (9.96) | 0; 41 |

Notes:

SD = Standard Deviation; POC = People of Color; CES-D = Centers for Epidemiology Scale – Depression

**Table 2.** LNPA engagement and adherence indicator descriptive statistics (n = 339).

| Indicator Variables | Mean (SD) | Median | Min; Max |
| --- | --- | --- | --- |
| **Engagement** | | | |
| Sum logins | 106.04 (182.59) | 39 | 1; 1362 |
| Sum page-hits | 321.15 (559.69) | 114 | 0; 4714 |
| Sum goals set | 25.05 (23.72) | 18 | 0; 81 |
| Sum forum posts | 1.81 (4.98) | 0 | 0; 36 |
| **Adherence** | | | |
| Sum weeks physical activity goal met | 9.85 (10.29) | 5 | 0; 27 |
| Sum complete diet days logged | 55.59 (63.20) | 27 | 0; 182 |
| Sum weight days logged | 91.51 (110.64) | 38 | 0; 379 |

Notes:SD = Standard Deviation

both indicate good fit. Additionally, the pooled CFIs for engagement = 0.971 and adherence = 0.974, as well as the pooled TLIs for engagement = 0.949 and adherence = 0.943, which all indicate good model fit for the data. These results are re-summarized in Table 3.

There were no observed direct effects between baseline depression symptoms and 6-month weight change in either of the engagement and adherence models (p's ≥ 0.6). However, results indicate that a one standard deviation increase in baseline CES-D scores was negatively associated with engagement ($\gamma_{eng,dep}$ = -0.315; SE = 0.082; p < 0.001) and with adherence ($\gamma_{adh,dep}$ = -0.333; SE = 0.083; p < 0.001), net of demographic covariates. In turn, these constructs were each associated with significant reductions in weight at 6 months, net of demographic covariates. As program engagement increased, participants lost an additional 2.1 kg by 6-months ($\gamma_{wtch,eng}$ = -2.097; SE = 0.209; p < 0.001), controlling for covariates. Further, as adherence to program recommendations increased, participants lost an additional 1.8 kg by 6-months ($\gamma_{wtch,adh}$ = -1.787; SE = 0.203; p < 0.001), controlling for covariates. Fig 2 displays the same path diagram with coefficients and factor loadings superimposed over directional arrows to aid interpretation.

While the primary outcomes of these models show how depressive symptoms can negatively influence engagement and adherence to then contribute to poorer weight change outcomes, there were several notable effects from the demographic control covariates: Age was the only consistently significant covariate across both engagement and adherence models, with increasing participant age above the program mean (51.86) showing positive effects on engagement ($\gamma_{eng,age}$ = 0.037; SE = 0.008) and adherence ($\gamma_{adh,age}$ = 0.028; SE = 0.008), and negative effects on 6-month weight change in the engagement ($\gamma_{wtch,age}$ = -0.057; SE = 0.025) and adherence ($\gamma_{adh,age}$ = -0.083; SE = 0.026) models (all p's ≤ 0.001). Participant sex was significantly associated with 6-month weight loss in the engagement model ($\gamma_{wtch,sex}$ = -1.667; SE

**Table 3.** SEM Fit Statistic Summary.

| Indicator | Value (p-value) | |
| --- | --- | --- |
| | **Engagement** | **Adherence** |
| Chi Square ($\chi^2$) | 39.58 (p = 0.001) | 37.066 (p < 0.001) |
| RMSEA | 0.063 (p = 0.189) | 0.089 (p = 0.016) |
| CFI | 0.971 | 0.974 |
| TLI | 0.949 | 0.943 |
| SRMR | 0.033 | 0.023 |

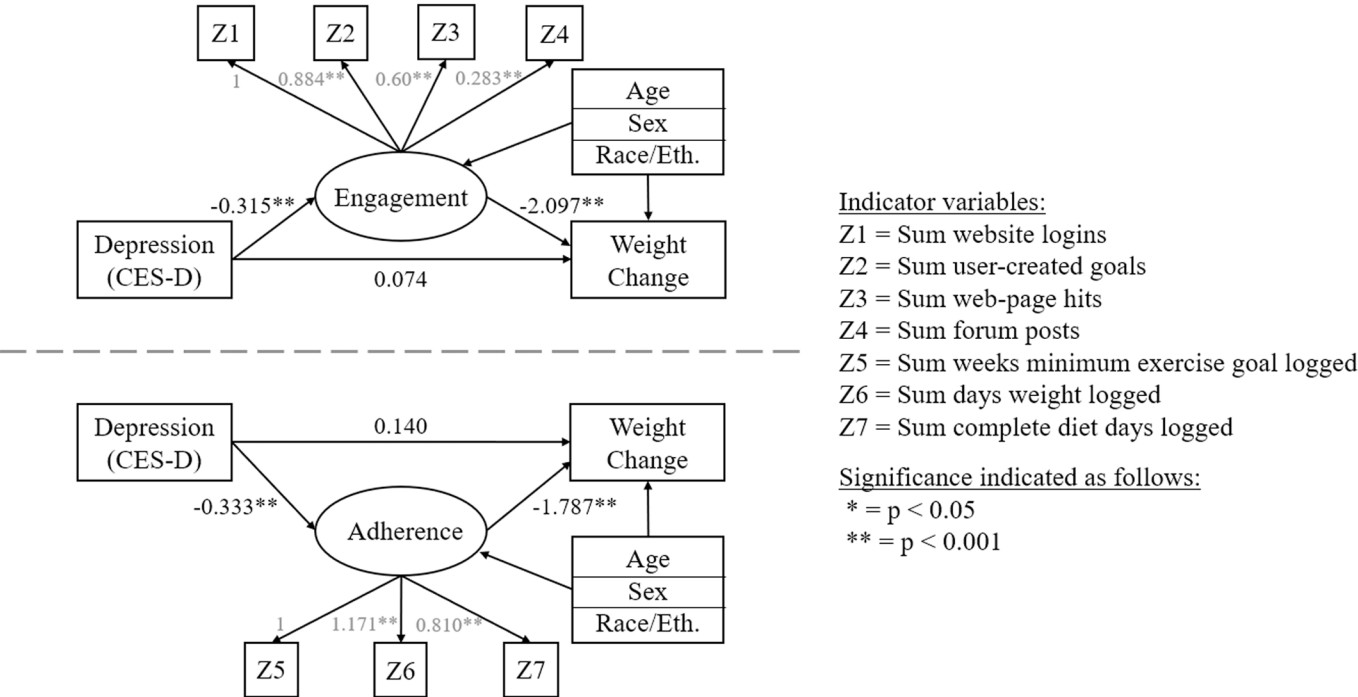

**Fig 2. Path diagrams with coefficients.** These results showed that the effects of baseline depression symptoms on weight change were completely mediated through the constructs of engagement and adherence in each model. The indirect effect of CES-D scores on 6-month weight change through engagement was 0.660 (SE = 0.170; $p < 0.001$). Rephrased in practical terms, participants with CES-D scores one standard-deviation above the mean (~23) engaged with the LNPA website less, and lost approximately 0.66% less weight by 6 months ($p < 0.01$) than those with average CES-D scores (13.13). Likewise, the indirect effect of CES-D scores on 6-month weight change through adherence was 0.594 (SE = 0.148; $p < 0.001$). Again rephrased in practical terms, participants with CES-D scores one standard deviation above the sample mean (~23) were less adherent to program self-monitoring and physical activity recommendations, and lost approximately 0.59% less weight by 6 months ($p < 0.01$) than those with average CES-D scores (13.13). These positive indirect effects on weight change would indicate attenuated weight loss, or possible weight gain, over 6 months by affected participants. Results did not meaningfully differ when using 12-month weight change values as the outcome.

= 0.588; $p = 0.005$) and barely reached significance in the adherence model ($\gamma_{wtch,sex} = -1.190$; SE = 0.604; $p = 0.049$), such that males tended to lose more weight than females in this study sample. Additionally, race/ethnicity was associated with reduced adherence only in that model ($\gamma_{adh,race} = -0.510$; SE = 0.224; $p = 0.023$). These results were largely replicated when using percent weight change as the outcome; however, sex was no longer a significant predictor of weight change. This suggests that while males may have lost more absolute weight in this study than females, the relative reductions in percent weight did not statistically differ.

## Discussion

The primary aim of this secondary analysis was to empirically test whether baseline depression symptoms exert undesirable influences on program engagement, adherence, and distal behavioral outcomes within a digital weight loss intervention. Results from this study fully support the guiding hypotheses and provide some quantitative estimates of the associations between depression symptoms and conduct within a digital weight management intervention. Results from this study also agree with previous research in traditional in-person weight management interventions that baseline depressive symptoms are negatively associated with intervention adherence, and thus contribute to undesirable change in weight outcomes [25].

This analysis shows that participants with mild to moderate depression symptoms (i.e., reporting CES-D scores < 16) can struggle to engage with website-based interventions and/ or to adhere to program recommendations, which can in turn have meaningful impacts on their behavior change outcomes. It is of growing importance for researchers to consider how depression may influence participant involvement in DBCIs, as population depression levels are climbing to unknown degrees since the COVID-19 pandemic, with some estimates documenting a >300% increase in adults reporting depression and anxiety symptoms between 2019-2021 [60,61]. National statistics tend to focus only on clinical, severe, or major depressive episodes, which likely underestimates the prevalence of people living with mild to moderate depression in the US [62,63].

This issue should be of particular interest for future digital weight management interventions, as it is known that depression symptoms and weight status are highly correlated, and that during the COVID-19 pandemic, those living with depression and anxiety experienced significantly greater weight gains than those without [64–66]. If these effects of depression symptoms remain unaddressed, future interventions may see reduced engagement and adherence, which can contribute to reduced weight change. As losing ≥5% body weight remains the clinically meaningful benchmark for weight loss [67], even small reductions in weight loss due to the influence of depression can impact a program's overall measured effectiveness among those participants most likely to need these tools. It would instead be prudent for future DBCIs to consider additional tailoring (i.e., personalizing intervention messaging and content) on depression symptoms and related factors measurable at baseline in tandem with sociodemographic and behavioral tailoring to potentially address these issues. Some possibilities might include tailoring the types of feedback messages sent by an intervention to avoid triggering negative attributions to failing goals [68,69], including lesson and review content tailored to depression and motivation at baseline to attempt to inoculate potential negative outcomes, relaxing goal standards in the early stages of an intervention to help participants to build initial self-efficacy, or possibly building in a 'time-out' feature if participants would want to take a short break from a program without worrying about failing goals in the interim. In a meta-synthesis of user experiences with DBCI's for depression and anxiety, Knowles *et al.* synthesized themes that computerized tailoring should be sensitive to 'Who I am' for personalized relevance, as well as to 'How I feel' to be appropriate for those experiencing low moods and low motivation typical of depression [70].

This analysis has several limitations. First, due to the nature of data available, there were limited system usage indicators to measure program adherence and engagement. While the original intent of this analysis was to model both engagement and adherence simultaneously, the two LVs were too highly correlated to model meaningful results, despite efforts to distinguish the page-hits indicator from all self-monitoring activities. This is likely because all behavioral self-monitoring in LNPA required manual logging using the IWL website. DBCIs using passive data collection methods such as wearable activity trackers to measure participant adherence to exercise goals may be more capable of comparing these LVs within the same model. Second, the LNPA study functionality which allowed participants to change dietary monitoring strategies and physical activity goal intensity *ad libitum* during the intervention changed the standards for how data were recorded (e.g., one dietary format had users manually log all calories they consumed, while another was assigned to a 'meal plan' format that automatically recorded preset menu items with calories and requested users note any deviations from that plan) and created challenges for consistently operationalizing longitudinal indicators of program adherence. This resulted in dietary and physical activity adherence indicators varying over time for a large number of participants and necessitated for all indicators to be brought into the same coarse scale for this analysis. This also may have contributed to

the adherence LV contributing less unique predictive capability than anticipated, as participants likely to be recorded as having higher adherence would necessarily need to log into the IWL website more often. Future examinations these relationships using longitudinal methods and measures will be valuable to understand if and how these relationships may vary over time. Third, the LNPA study sample lacked racial and gender diversity with a majority proportion of white female participants, which is unfortunately not uncommon in DBCI research [47,48].

Strengths of this analysis include the high-fidelity, automated program delivery across all users within a real-world setting and extensive collection of objective participant usage metrics. These manifest, objective indicators enabled this detailed analysis of how baseline depressive symptoms likely influenced how participants used this intervention, and the resulting undesirable impacts on weight loss, without relying on self-report measures which could be biased. Additionally, the parent study included a majority of participants who had not achieved a Bachelor's degree or equivalent, approximately 25% male participants, included many older adult participants with a wide range of BMIs, and also included many adults living with comorbidities such as diabetes all recruited within primary care settings. Thus, the results and conclusions from this study are likely be somewhat more generalizable to participants recruited for DBCIs within a primary care setting.

## Conclusion

This secondary analysis adds a novel contribution to the DBCI literature as being among the first to empirically show that depression symptoms can negatively influence distal weight loss within a website-based digital weight management intervention, and that this effect is fully mediated through the latent constructs of engagement and adherence. This finding may point to an important factor contributing to observed declines in participant engagement over time in DBCIs. The field of digital behavior change interventions would greatly benefit from future research examining if tailoring digital interventions on baseline depression levels can help ameliorate these negative relationships, and whether other psychological factors may be influencing participant engagement, adherence, and overall success in these interventions.

## Supporting information

**S1 File. Structural Equation Model Indicator Correlation Matrix.**
(CSV)

## Acknowledgments

The authors would like to thank the faculty and staff of the UNC Weight Research Program.

## Author contributions

**Conceptualization:** Lex Hurley.

**Data curation:** Lex Hurley.

**Formal analysis:** Lex Hurley.

**Funding acquisition:** Deborah F. Tate.

**Methodology:** Lex Hurley, Nisha G. O'Shea.

**Supervision:** Nisha G. O'Shea, Deborah F. Tate.

**Visualization:** Lex Hurley.

**Writing – original draft:** Lex Hurley.

**Writing – review & editing:** Lex Hurley, Nisha G. O'Shea, Julianne Power, Christopher Sciamanna, Deborah F. Tate.

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
