## [Decision Letter · Decision Letter 0]

23 Sep 2024

PDIG-D-24-00324

Measuring the influence of depressive symptoms on engagement, adherence, and weight loss in a 6-month eHealth intervention

PLOS Digital Health

Dear Dr. Hurley,

Thank you for submitting your manuscript to PLOS Digital Health. After careful consideration, we feel that it has merit but does not fully meet PLOS Digital Health's publication criteria as it currently stands. Therefore, we invite you to submit a revised version of the manuscript that addresses the points raised during the review process.

Please submit your revised manuscript within 60 days Nov 22 2024 11:59PM. If you will need more time than this to complete your revisions, please reply to this message or contact the journal office at digitalhealth@plos.org. Please include the following items when submitting your revised manuscript:

We look forward to receiving your revised manuscript.

Kind regards,

Haleh Ayatollahi

Section Editor

PLOS Digital Health

Journal Requirements:

1. Your current Financial Disclosure states, “T32 Cancer Health Disparities Training Grant from the National Cancer Institute of the National Institutes of Health”. However, your funding information on the submission form indicates that you received funding from “Center for Biomedical Informatics and Information Technology, National Cancer Institute” and Grant Recipient "Dr. Lex Hurley". Please indicate by return email the full and correct funding information for your study and confirm the order in which funding contributions should appear. Please be sure to indicate whether the funders played any role in the study design, data collection and analysis, decision to publish, or preparation of the manuscript.

2. In the online submission form, you indicated that "The authors do not have permission to share raw study data due to requirements to protect the privacy of participants, in accordance with their informed consent. However, de-identified data used in this specific analysis and/or output files related to this analysis will be made available upon request to the corresponding author.". 

3. Uploaded as supplementary information.

3. We notice that your figures are included in the manuscript file. Please remove them as there is already Figures uploaded in the file inventory. Also remove the citation "Supporting information S1 Fig. Structural equation model path diagrams S2 Fig. Path diagrams with coefficients" listed in the manuscript after the references list.

Additional Editor Comments (if provided):

Reviewers' comments:

Reviewer's Responses to Questions

**Comments to the Author**

1. Does this manuscript meet PLOS Digital Health’s publication criteria ? Is the manuscript technically sound, and do the data support the conclusions? The manuscript must describe methodologically and ethically rigorous research with conclusions that are appropriately drawn based on the data presented.

Reviewer #1: Partly

Reviewer #2: Yes

Reviewer #3: No

2. Has the statistical analysis been performed appropriately and rigorously?

Reviewer #1: N/A

Reviewer #2: Yes

Reviewer #3: I don't know

3. Have the authors made all data underlying the findings in their manuscript fully available (please refer to the Data Availability Statement at the start of the manuscript PDF file)?

Reviewer #1: No

Reviewer #2: No

Reviewer #3: Yes

4. Is the manuscript presented in an intelligible fashion and written in standard English?

PLOS Digital Health does not copyedit accepted manuscripts, so the language in submitted articles must be clear, correct, and unambiguous. Any typographical or grammatical errors should be corrected at revision, so please note any specific errors here.

Reviewer #1: No

Reviewer #2: Yes

Reviewer #3: No

5. Review Comments to the Author

Please use the space provided to explain your answers to the questions above. You may also include additional comments for the author, including concerns about dual publication, research ethics, or publication ethics. (Please upload your review as an attachment if it exceeds 20,000 characters)

Reviewer #1: Dear Authors, 

Your research has not found any new cofounder of depression. Weight gain due to depression and vice versa are the known facts now. 

Therefore, I am rejecting it for publication. 

Thank you.

Reviewer #2: Review of Manuscript: "Measuring the influence of depressive symptoms on engagement, adherence, and weight loss in a 6-month eHealth intervention"

This manuscript addresses a pertinent issue in the field of digital health, specifically the influence of depressive symptoms on engagement, adherence, and subsequent weight loss in an eHealth intervention. Given the increasing prevalence of digital behavior change interventions for weight management, understanding the psychological factors that impact their effectiveness is both timely and relevant. The use of structural equation modeling (SEM) to explore the relationships between depression, engagement, adherence, and weight outcomes is a strength of the study, as it allows for a nuanced understanding of these complex interactions. However, there are areas where clarity, methodological detail, and interpretation of results could be improved to enhance the manuscript's overall contribution to the field.

Abstract: 

The abstract does address the core components (background, methods, results, discussion), but the methods section could use more detail to fully explain the research approach. The results section lacks quantitative results (e.g., statistical significance values), which are often expected in scientific abstracts. For instance, line 36, the phrase “positive indirect association” might confuse some readers without further clarification that it leads to less weight loss or potential weight gain. The discussion provides a broad statement but lacks actionable insights or suggestions, which could strengthen the conclusion.

Introduction:

The introduction provides a solid background, defining key concepts and linking them to previous research. However, the review of literature might benefit from being a bit more focused, as the numerous references to different studies could overwhelm readers without clear synthesis. The hypotheses are clearly stated at the end of the introduction, outlining the goals of the analysis. This sets the stage for the research question and the methodology used to address it.

Suggestions: The 'Law of Attrition' is referenced as an important concept in DBCI research, but a brief explanation of why this law is relevant to the current study (beyond novelty) might strengthen the argument.

The discussion around the possible role of depression as a contributor to disengagement/dropout could be expanded to explain why this is the case (e.g., motivation, fatigue, psychological barriers).

Some of the references to past studies could be more tightly integrated into the central narrative to reduce the feeling of listing various sources. It would be beneficial to show how they collectively lead to the formulation of the study's hypotheses.

Methods: 

The Methods section provides a thorough and structured description of the study's design, dataset, measures, and analytical approach, making it very clear for replication or further understanding of the research process. Overall, the methods are strong, comprehensive, and appropriate for the study objectives.

Results: 

The section begins by clearly explaining the fit of the simultaneous equation models (SEM) using various fit indices (RMSEA, CFI, TLI, SRMR), which is a good approach. The use of multiple indices provides a comprehensive picture of model performance. It could be improved by briefly explaining why chi-square (χ²) was significant despite the other fit indices indicating good fit. A more nuanced discussion of the limitations of χ² with small sample sizes could enhance the reader’s understanding. 

The statistical results are well-presented with relevant coefficients, standard errors (SE), and p-values, providing transparency. The text describes the absence of direct effects and highlights the mediation role of engagement and adherence in the relationship between depression symptoms and weight change, which is critical. Consider adding a more intuitive explanation of the indirect effects. For instance, explaining how engagement and adherence mediate weight change could be framed more clearly to emphasize the importance of these constructs.

The key results are summarized efficiently in Table 3, and the significant effects of age, sex, and race/ethnicity on engagement, adherence, and weight change are appropriately addressed. The section could benefit from clearer transitions between the main results (e.g., engagement/adherence) and the secondary findings (e.g., demographic effects). This would help maintain focus on the primary hypotheses and findings while still reporting on important covariates.

The indirect effects of CES-D scores on weight change are explained with practical examples, such as a standard deviation increase in CES-D scores leading to specific percentages of reduced weight loss. These practical interpretations are excellent, but the narrative could expand on why these indirect effects are meaningful in a real-world context. For example, explaining how even a modest decrease in weight loss due to lower engagement/adherence could have clinical implications might strengthen the impact of the findings.

Discussion: 

The discussion highlights key strengths, including the high-fidelity delivery and the objective collection of participant usage metrics, which are important aspects of the study. The strengths could be expanded by emphasizing the novelty or significance of the findings in digital health research. For instance, discussing how these insights add to the understanding of the impact of depression on digital health interventions would underscore the contribution of the study.

Reviewer #3: I think your research questions are very important hence, I didn't reject your draft although I have serious concerns about the fundamentals of methodology, introduction, and language of draft overall (e.g., full of jargon) that requires further work on clarification and correction. I would prefer to give you a second chance in writing the draft given that you were honest in sharing your limitations and have important research questions maybe you can do modifications or rectify the problems or add more information and resubmit the draft. If you have omitted information that may have led to us not understanding something in your study, this is your chance to clarify and add that to improve your draft. I highly recommend that you consult with the clinical psychologist in your team about the psychological constructs concern for example, she could have helpful insights. Please find my specific comments on the attached pdf file for your review. Hope they are helpful.

6. PLOS authors have the option to publish the peer review history of their article (what does this mean? ). If published, this will include your full peer review and any attached files.

**Do you want your identity to be public for this peer review?** For information about this choice, including consent withdrawal, please see our Privacy Policy .

Reviewer #1: Yes: Subhagata Chattopadhyay

Reviewer #2: No

Reviewer #3: No

---

## [Decision Letter · Decision Letter 1]

21 Nov 2024

PDIG-D-24-00324R1Measuring the influence of depressive symptoms on engagement, adherence, and weight loss in an eHealth interventionPLOS Digital Health Dear Dr. Hurley, Thank you for submitting your manuscript to PLOS Digital Health. After careful consideration, we feel that it has merit but does not fully meet PLOS Digital Health's publication criteria as it currently stands. Therefore, we invite you to submit a revised version of the manuscript that addresses the points raised during the review process. Please submit your revised manuscript within 60 days Jan 20 2025 11:59PM. If you will need more time than this to complete your revisions, please reply to this message or contact the journal office at digitalhealth@plos.org. Please include the following items when submitting your revised manuscript:* A rebuttal letter that responds to each point raised by the editor and reviewer(s). You should upload this letter as a separate file labeled 'Response to Reviewers '. This file does not need to include responses to any formatting updates and technical items listed in the 'Journal Requirements' section below.* A marked-up copy of your manuscript that highlights changes made to the original version. You should upload this as a separate file labeled 'Revised Manuscript with Track Changes '.* An unmarked version of your revised paper without tracked changes. You should upload this as a separate file labeled 'Manuscript '. If you would like to make changes to your financial disclosure, competing interests statement, or data availability statement, please make these updates within the submission form at the time of resubmission. Guidelines for resubmitting your figure files are available below the reviewer comments at the end of this letter. We look forward to receiving your revised manuscript. Kind regards, Haleh AyatollahiSection EditorPLOS Digital Health Leo Anthony CeliEditor-in-ChiefPLOS Digital Healthorcid.org/0000-0001-6712-6626 **Journal Requirements:** **Additional Editor Comments (if provided):****Reviewers' Comments:** Reviewer's Responses to Questions

**Comments to the Author**

1. If the authors have adequately addressed your comments raised in a previous round of review and you feel that this manuscript is now acceptable for publication, you may indicate that here to bypass the “Comments to the Author” section, enter your conflict of interest statement in the “Confidential to Editor” section, and submit your "Accept" recommendation.

Reviewer #3: (No Response)

Reviewer #4: All comments have been addressed

Reviewer #5: (No Response)

2. Does this manuscript meet PLOS Digital Health’s publication criteria ? Is the manuscript technically sound, and do the data support the conclusions? The manuscript must describe methodologically and ethically rigorous research with conclusions that are appropriately drawn based on the data presented.

Reviewer #3: No

Reviewer #4: Yes

Reviewer #5: Partly

3. Has the statistical analysis been performed appropriately and rigorously?

Reviewer #3: I don't know

Reviewer #4: Yes

Reviewer #5: I don't know

4. Have the authors made all data underlying the findings in their manuscript fully available (please refer to the Data Availability Statement at the start of the manuscript PDF file)?

Reviewer #3: (No Response)

Reviewer #4: Yes

Reviewer #5: Yes

5. Is the manuscript presented in an intelligible fashion and written in standard English?

Reviewer #3: Yes

Reviewer #4: Yes

Reviewer #5: Yes

6. Review Comments to the Author

Reviewer #3: (No Response)

Reviewer #4: I am satisfied with the revisions made.

Reviewer #5: I am reviewing the manuscript “PDIG-D-24-00324_R1_reviewer” where the authors examine how depressive symptoms are associated with engagement and outcome of an eHealth weight-loss intervention.

The study makes a clear contribution to literature. However, I consider the made conceptual and empirical differentiation between “engagement” and “adherence” lacks precision and requires revision.

Major comments:

• As the authors comment, the existing methods for defining and measuring engagement and adherence vary considerably. Here, I find that the two are conceptualized in a way that does not allow differentiating them and contributes to the confusion.

• Conceptually, the authors describe that engagement is “meaningful use”. Adherence is conceptualized as “behavior corresponding with intervention goals”. Are these two definitions not intrinsically related? It appears that adherence is compliance with intervention requests, but these requests should be indications of meaningful use. Thus, if adherence is meeting “meaningful use” goals as defined by researchers, the two concepts are overlapping, which challenges differentiating between them.

• Methodologically, the conceptualization of engagement includes website logins while adherence includes e.g. exercise goals logged. Here, doing the latter requires doing the former, which makes the two concepts naturally linked and correlated.

• Thus, the used conceptual and methodological definitions for “engagement” and “adherence” are interrelated, which confuses the reader. I suggest using either concept, or considerably clarifying on the distinction between “engagement” and “adherence”. This revision should include includes both the “Introduction” section, and “Methods” where the concepts are introduced and then operationalized. Some solutions include conceptualizing engagement as intervention use (omitting the concept of adherence) or showing how engagement is precondition for adherence.

• The intercorrelations for Z1-Z7 are now not described. The reader would benefit from such information, which would also rationalize how the constructs that combine these metrics were created. Are the within-group correlations (Z1-Z4 and Z5-Z7) lower/higher/equal than the correlations between groups of Engagement-Adherence?

Minor comments

• “Approximately 53 participants 266 logged in 5 times or fewer over 6 months”, please add percentage after the “53”.

• “Age was the only consistently significant covariate across both engagement and adherence models, showing positive effects on engagement” in this section, please clarify whether you mean that higher age is associated with higher engagement for instance. Consider clarifying the other statements as well.

• The study should report the nationality of the sample or location of where the sample was drawn from.

• The reader would benefit from a Figure that illustrates the intervention in the section “Dataset”.

• I suggest revising the headers. Is “Materials and methods” simply “Methods”? Could the “Dataset” be divided into “Intervention” or “Study procedures” section? Should there be “Participants” section?

• In Figure 2, are the correlations truly “1” for Engagement-Z1, and Adherence-Z5? If they are, they should be marked with asterisks? Two decimals should be enough precision. Also describe what r stands for.

7. PLOS authors have the option to publish the peer review history of their article (what does this mean? ). If published, this will include your full peer review and any attached files.

**Do you want your identity to be public for this peer review?** For information about this choice, including consent withdrawal, please see our Privacy Policy .

Reviewer #3: No

Reviewer #4: No

Reviewer #5: **Yes: ** Lauri Lukka

---

## [Decision Letter · Decision Letter 2]

24 Jan 2025

Measuring the influence of depressive symptoms on engagement, adherence, and weight loss in an eHealth intervention

PDIG-D-24-00324R2

Dear Dr. Hurley,

We are pleased to inform you that your manuscript 'Measuring the influence of depressive symptoms on engagement, adherence, and weight loss in an eHealth intervention' has been provisionally accepted for publication in PLOS Digital Health.

Best regards,

Haleh Ayatollahi

Section Editor

PLOS Digital Health

**Additional Editor Comments (if provided):**

**Reviewer Comments (if any, and for reference):**

Reviewer's Responses to Questions

**Comments to the Author**

1. If the authors have adequately addressed your comments raised in a previous round of review and you feel that this manuscript is now acceptable for publication, you may indicate that here to bypass the “Comments to the Author” section, enter your conflict of interest statement in the “Confidential to Editor” section, and submit your "Accept" recommendation.

Reviewer #4: All comments have been addressed

Reviewer #5: All comments have been addressed

2. Does this manuscript meet PLOS Digital Health’s publication criteria ? Is the manuscript technically sound, and do the data support the conclusions? The manuscript must describe methodologically and ethically rigorous research with conclusions that are appropriately drawn based on the data presented.

Reviewer #4: Yes

Reviewer #5: Yes

3. Has the statistical analysis been performed appropriately and rigorously?

Reviewer #4: Yes

Reviewer #5: Yes

4. Have the authors made all data underlying the findings in their manuscript fully available (please refer to the Data Availability Statement at the start of the manuscript PDF file)?

Reviewer #4: Yes

Reviewer #5: Yes

5. Is the manuscript presented in an intelligible fashion and written in standard English?

Reviewer #4: Yes

Reviewer #5: Yes

6. Review Comments to the Author

Reviewer #4: I am happy with the revisions made

Reviewer #5: I thank the Authors for comprehensively addressing the comments I made in my review.

7. PLOS authors have the option to publish the peer review history of their article (what does this mean? ). If published, this will include your full peer review and any attached files.

**Do you want your identity to be public for this peer review?** For information about this choice, including consent withdrawal, please see our Privacy Policy .

Reviewer #4: No

Reviewer #5: **Yes: ** Lauri Lukka
